# HIV patients' perceptions of a potential multi-component mindfulness-based smoking cessation smartphone application intervention

Taghrid Asfar[1,2]*, Maria Luisa Alcaide[3,4], Deborah L. Jones[5], Laura A. McClure[1], Judson Brewer[6], David J. Lee[1,2], Adam Carrico[1]

1 Department of Public Health Sciences, University of Miami Miller School of Medicine, Miami, FL, United States of America, 2 Sylvester Comprehensive Cancer Center, University of Miami Miller School of Medicine, Miami, FL, United States of America, 3 Division of Infectious Diseases, Department of Medicine, University of Miami Miller School of Medicine, Miami, FL, United States of America, 4 Internal Medicine, Jackson Memorial Hospital, Miami, FL, United States of America, 5 Department of Psychiatry & Behavioral Sciences, University of Miami Miller School of Medicine, Miami, FL, United States of America, 6 Department of Behavioral and Social Sciences, Brown Mindfulness Center, Brown University School of Public Health, Providence, RI, United States of America

* tasfar@miami.edu

**Data Availability Statement:** All relevant data are within the paper and its Supporting Information files.

## Abstract

### Objectives

Cigarette smoking rates among people living with HIV (PLWH) in the US is triple that of the general population. PLWH smokers are a high-risk group for smoking-related health disparities and should be a prime focus for smoking cessation efforts. Our team has developed a novel evidence-based Mindfulness Training (MT) smoking cessation smartphone application (app), "Craving-to-Quit." Using qualitative focus groups among PLWH smokers, this study aims to tailor and optimize the app's content and design to PLWH's unique psychosocial profile and needs.

### Methods

We conducted 8 focus groups among PLWH smokers (n = 59; 47.5% females; ≥18 years) to gain insight into participants' perceptions about the app, MT, and the feasibility and acceptability of adding two additional strategies (CM: Contingency Management; self-monitoring of anti-retroviral therapies intake [ART]) to further optimize the app. Participants were asked to practice MTs and watch videos from the app presented on a screen in the conference room to discuss their experience. Sessions were audio-taped, transcribed verbatim, and analyzed thematically using NVivo.

### Results

Most participants were non-Hispanic black (67.8%), on a federal health insurance program (61.0%). Participants considered it easy to learn the app and thought that MT is helpful in reducing stress and motivating quit attempts and were supportive of adding CM and

**Funding:** Funding for this study was provided by the Miami Center for AIDS Research (CFAR) at the University of Miami (P30AI07396 - CFAR developmental award; P30MH116867). Supplemental funding was provided by the Sylvester Comprehensive Cancer Center at the University of Miami Miller School of Medicine. Research reported in this publication was also supported by the National Cancer Institute of the National Institutes of Health under Award Number P30CA240139. The content is solely the responsibility of the authors and does not necessarily represent the official views of the National Institutes of Health.

**Competing interests:** The authors have declared that no competing interests exist.

recommended providing $20-$50 weekly cash incentives to help in quitting. Participants felt that adding self-monitoring of ART is helpful but were concerned about confidentiality in case they lost their phone. Participants recommended making the app cost-free and adding information about smoking cessation medications and the negative effects of smoking among PLWH.

## Conclusions

Findings will guide the development of a novel multi-component smoking cessation intervention app integrating MT, CM, and ART self-monitoring strategies. This intervention has the potential to address several barriers to quitting in PLWH. Further clinical research is needed to test this intervention.

## Introduction

People living with HIV (PLWH) in the US are at high risk for tobacco-related health disparities. Cigarette smoking prevalence among PLWH is triple that of the general population (47%–65% vs. 17%, respectively) [1, 2]. Most PLWH who smoke are members of marginalized groups (e.g., ethnic minorities, migrants, men who have sex with men) [3], unemployed, and have low social support [4]. Compared to nonsmokers, PLWH who smoke have threefold the risk of cancers [5], double the risk of cardiovascular complications [6], a 6 to 15 years shorter lifespan [7], and lower self-reported quality of life [8]. Yet despite their high interest in quitting (40%–75%) [9], PLWH who smoke lack access to cessation treatment in clinical settings [3, 10, 11]. Research suggests that PLWH who smoke are less likely to be advised by healthcare providers to quit smoking and to attend smoking cessation counseling if they are referred (38% and 3%, respectively) [10, 11]. In addition, PLWH have a complex psychosocial profile that hinders their smoking cessation efforts [3, 12]. Mainly, depression is a significant barrier to quitting smoking among PLWH [13–15]. This is concerning given that 40%–60% of PLWH are depressed [16], a rate three times that of the general population (3%–14%) [17, 18]. Furthermore, cigarette smoking among PLWH is associated with lower adherence to anti-retroviral therapies (ART), which can influence HIV pathogenesis [19–21]. Therefore, PLWH who smoke are a high-risk group for smoking-related health disparities and should be a prime focus for smoking cessation efforts. In particular, novel smoking cessation interventions that can improve their access to cessation treatment and address their unique psychosocial profile are urgently needed.

A promising method to improve PLWH's access to smoking cessation treatments is to deliver the intervention by smartphone application (app). Smartphone-based apps have emerged as important tools for health-related behavioral interventions. Two systematic reviews concluded that smartphone-based smoking cessation apps are effective and significantly increase access to treatment [22, 23]. Compared to in-person treatment, this approach can be standardized, reduce the stigma associated with seeking treatment, allow the use of multiple methods to deliver the intervention (e.g., video, audio), facilitate the integration of the treatment into the user's daily life, and simultaneously boost user engagement, a strong predictor of a successful smoking cessation [24–27]. Smartphone ownership among PLWH is comparable to the general population [28], and the growth in ownership is most pronounced in ethnic minority groups and those with low incomes, who constitute the majority of the PLWH

community in the US [25]. Therefore, smartphone app interventions may have a broad reach among PLWH and can address tobacco-related health disparities in this high-risk group [29].

*Mindfulness Training* (MT) smoking cessation interventions aim to increase an individual's awareness of their environment, thoughts, emotions, and physical sensations as related to cravings by helping participants cultivate the ability to "sit with" discomfort, which usually manifests via craving for a cigarette [30–32]. Several meta analyses and systematic reviews have indicated that MTs for smoking cessation are effective [33, 34]. In PLWH, MTs have been feasible and effective in improving quality of life, emotional well-being, immunological status, and coping with HIV [35, 36]. However, MT-based smoking cessation interventions have not been tested in PLWH. Another strategy that might improve smoking cessation outcomes among PLWH is *Contingency Management* (CM). CM is an evidence-based behavioral intervention in which individuals receive tangible reinforcement for biologically confirmed substance abstinence [37]. CM has been successful in retaining patients in treatment and fostering stable periods of abstinence in substance use behavioral research, including tobacco-use research [38, 39]. A recent Cochrane review concluded that there is high certainty evidence that CM strategies improve smoking cessation rates at long-term follow-up in mixed population studies [40]. In PLWH, CM has been feasible and effective in improving compliance to ART [41] and reducing several risk behaviors (e.g., unprotected sex, injection drug use) [42]. CM may also reduce the economic stressor for smoking cessation (e.g., unemployment) in PLWH [40, 43]. A pilot trial testing a smartphone-based MT intervention combined with CM in smokers with mood disorders showed a significant difference in abstinence rates in the intervention versus standard care at the end-of-treatment [44]. Therefore, CM may serve as an ideal adjunct intervention to MT to promote abstinence in PLWH who are seeking tobacco cessation treatment.

Finally, given the association between cigarette smoking and low adherence to ART among PLWH, integrating self-monitor strategies for ART intake in the smoking cessation treatment has the potential to improve PLWH's adherence to treatment. Based on Social Cognitive Theory, self-monitoring, the active observation and recording of behaviors, states, and their determinants and effects, is a core element of self-regulation and self-management because it helps patients identify triggers to noncompliance at the time and in the context in which they occur, and thus ultimately gain control over their behavior [45, 46]. Self-monitoring has been used broadly as a minimal-intervention strategy for a variety of behavioral modification efforts, and is typically used as an adherence intervention in clinical trials of medications [47]. Furthermore, a recent systematic review of interventions to improve self-management in PLWH concluded that technology-assisted (e.g., phone, website) self-monitoring interventions are effective in maintaining medication adherence in PLWH [48]. Thus, further investigation of the benefit of combining these methods with smoking cessation is warranted.

Our team has developed a novel evidence-based MT relapse prevention smoking cessation app entitled "Craving-to-Quit" [32, 49]. In this study, we conducted focus groups to adapt and optimize the content and design of the Craving-to-Quit app to PLWH's unique values and beliefs. Focus groups discussed the participants' perceptions of the app (e.g., value of app health technology, features, usability, videos, message content), MT (e.g., intensity, usefulness, challenges, recommendations for improvement), CM (e.g., feasibility, amount and type of incentives, schedule of payment), and adding strategies to monitor adherence to ART (feasibility, concern). Results will inform the adaptation and optimization of the content and design of a novel multi-component smoking cessation smartphone app that has the potential to address several barriers to quitting in PLWH who smoke.

## Materials and methods

### Design

The study was approved by the University of Miami (UM) Institutional Review Board (IRB ID: 20190547). Eight focus groups were conducted among PLWH who smoke (n = 59; 47.5% females; >18 years) from December 2019 to February 2020. Four groups were comprised of males, and four groups were comprised of females. Recruitment was done through purposive and snowball sampling using the HIV registry and Research Unit at UM. Eligibility criteria were >18 years, diagnosed with HIV, have smoked ≥ 5 cigarettes/day in the past year, interested in quitting smoking in the next 30 days, own a smartphone, read/speak English, able to provide informed consent, and willing to attend the focus group discussion as required. These criteria were chosen because they represented the eligibility criteria to participate in the planned smoking cessation trial that will test the developed intervention in the future. Focus groups took place in a private conference room at an academic institution, and each lasted 60–90 minutes. Focus group discussions were moderated by the study investigator and two public health graduate students with training in qualitative research. Group discussions were guided by a semi-structured script based on the *Tailored Health Communications* (THCs) Framework [50]. According to the THCs framework, tailoring could enhance motivation to process health information in at least four ways: (a) match content to an individual's information needs and interests, (b) frame health information in a context that is meaningful to the person, (c) use design and production elements to capture the individual's attention, and (d) provide information in the amount, type, and through channels of delivery preferred by the individual, thus potentially reducing barriers to exposure of individuals to communication interventions. Such an approach then could increase attention, lead to subsequent yielding, and ultimately enhance the likelihood of behavior change. The UM participants received a $100 incentive for participating in the focus groups.

### The "Craving-to-Quit" app

The "Craving-to-Quit" app includes 22 modules for 22 days, 5–15 minutes each day, designed to teach MT using audio, video, and animated lessons (Table 1) [51]. Participants have access to only one new module per day, and subsequent days are locked to prevent skipping ahead. A quit date is scheduled on day 21 in the app. The app teaches three basic formal MT techniques including body scan (bringing awareness to different parts of the body to foster awareness of body sensations that constitute cravings and affective states), loving kindness (repeating phrases such as "may X be happy" to foster acceptance of oneself and others), and breath awareness (paying attention to the breath wherever one feels it most strongly in the body to help retrain the mind away from habitually engaging in self-related thinking and toward a more present-centered awareness), and one informal MT called RAIN (Recognize, Accept, Investigate, and Note what cravings feel like as they arise/pass away). The app also includes other features such as social support (quit friend sign-ups, the tip of the week), activity feed (to track interactions with the app), and "my morning stats" (to track smoking status every day in the morning).

### Participants and recruitment

Potential participants were identified through the HIV registry and Research Unit at UM. Participants who consented to be contacted and identified as smokers were contacted by phone or in person to invite them to participate. Participants were also asked to refer eligible peers. Those who were interested in participation were screened for eligibility and scheduled for focus group

**Table 1. The content of the Craving-to-Quit mindfulness app.**

| Week (1) | • **Day 1:** Introduces the Craving-to-Quit app, mindfulness, habit formation, and mindful smoking exercise.<br>• **Day 2:** Asks to set personalized goals and provides a mindful smoking exercise.<br>• **Day 3:** Teaches body scan meditation (bringing awareness to different parts of the body to foster awareness of body sensations that constitute cravings and affective states) and provides a mindful smoking exercise.<br>• **Day 4:** Teaches how to work with cues, affective states, and craving using RAIN (Recognize, Accept, Investigate, and Note what cravings feel like as they arise/pass away), and provides a RAIN exercise. In RAIN, participants are asked to identify their smoking trigger, rate their craving, and choose between using RAIN to ride out their craving, or completing an audio-guided exercise to "smoke mindfully" by paying attention to the moment-to-moment experience and bodily sensations of smoking.<br>• **Day 5:** Introduces the concept of craving using an animation with the metaphor of craving as a tantrum toddler, i.e., let the toddler cry it out, and provides a RAIN exercise.<br>• **Day 6:** Teaches how to recognize triggers and provides a RAIN exercise.<br>• **Day 7:** Expands on the concept of craving using an animation with the metaphor of craving as a fire and provides a RAIN exercise. |
|---|---|
| Week (2) | • **Day 8:** Teaches how to use noting practice, i.e., the "N" of RAIN, in everyday life, and provides a noting practice exercise.<br>• **Day 9:** Teaches strategies for staying on track and provides a noting exercise.<br>• **Day 10:** Teaches resistance training and provides a noting exercise.<br>• **Day 11:** Builds on noting practice by teaching curiosity, a core element of mindfulness, and provides a curiosity exercise.<br>• **Day 12:** Expands on the concept of craving and curiosity using an animation with the metaphor of a hot coal, asks "What do you get from smoking mindfully today?"<br>• **Day 13:** Teaches loving-kindness meditation (repeating phrases such as "may X be happy" to foster acceptance of oneself and others), provides a loving-kindness exercise, and provides "Wild Geese" poem by Mary Oliver.<br>• **Day 14:** Teaches evaluating the costs & benefits of smoking, provides a loving-kindness exercise. |
| Week (3) | • **Day 15:** Discusses misperceptions about quitting and how to get social support.<br>• **Day 16:** Builds on noting and curiosity by teaching noting while walking meditation, provides a walking noting practice.<br>• **Day 17:** Teaches open awareness of thoughts, to work mindfully with thoughts that trigger smoking, using animations such as "Thoughts like a Radio."<br>• **Day 18:** Builds on walking while noting with animations such as "Tripping on Thoughts," "Autobiography in 5 short chapters" reading by Portia Nelson, provides a noting exercise.<br>• **Day 19:** Asks to reflect on experience with treatment, noting practice with a particular eye out for doubt.<br>• **Day 20:** Provides tips on staying motivated and maintaining mindfulness practice, writes a mantra to use and sets mantra reminder.<br>• **Day 21:** Quit day ceremony, tell a friend/family that today is their quit day. |
| Week (4) | • **Day 22:** Incorporates mindfulness practices as a new, healthy habit, and instructs the user on which modules to return to if they relapse.<br>• Bonus "Big Mind Meditation" audio by Joseph Goldstein; Tree Analogy for reinforcing noting video; Attitude is Everything video; "Mountain Meditation" audio by Joseph Goldstein; Sitting Meditation audio. |

discussion. Overall, 67 participants were screened, 1 was ineligible, 2 declined participation, 5 were scheduled but did not show up for the discussion session, and 59 completed the study.

## Procedures

Each session began with a general discussion on the nature, confidentiality, and general interaction preferences for the group discussion. After explaining the study and obtaining written informed consent, participants completed a baseline assessment followed by the focus group discussion. The baseline assessment collected information on demographic characteristics (age, gender, race/ethnicity, educational level, employment status, income), smoking history, Fagerström Test for Nicotine Dependence (FTND; high dependence if the score > 6) [52], self-efficacy [53], alcohol use (ASSIST) [54], the visual analogue scale (VSA) of adherence to

ART (suboptimal adherence is defined as reporting <90% adherence to ART in the past 30 days) [55], and depression (CES-D; depressed if the score $\geq$ 10) [56]. Breath samples were collected using a hand-held CO monitor (Bedfont piCO+Smokerlyzer; Bedfont Scientific Ltd, Maidstone) to validate participants' smoking status (CO $\geq$ 6 ppm is the cut-off point for being an active smoker) [57]. A semi-structured focus group guide developed by the research study team explored the following topics: 1) the perceived barriers to smoking cessation, 2) views on the Craving-to-Quit app (features, usability/usefulness, message content), 3) views on MT (usefulness, intensity, challenges, improvement), 4) views on CM (feasibility, amount and type of incentives, schedule of payment), 5) views on adding strategies to monitor adherence to ART (usefulness, relevance, concerns), and 6) recommendations for improving the app. Given that participants did not have the chance to undergo the full 22-day intervention, they received a video demonstration explaining the app content and features, and then they were given the opportunity to try the features within the focus group before providing their feedback. Participants were also asked to practice MT exercises and watch videos from the app, followed by discussion about their experience.

## Analysis

Statistical analysis of the baseline assessment was conducted using SAS Software v9.4 (SAS Institute Inc., Cary, NC). For the baseline assessment, we calculated frequencies and percentages for categorical variables and mean and standard deviation for continuous variables (Tables 2 & 3). Focus group data were audio-recorded, transcribed, and coded in NVivo 12 Software [58]. We qualitatively analyzed the data using techniques that incorporated the stages of familiarization, identification of a thematic framework, indexing, charting, mapping, and interpretation [59]. Coding of participants' perceptions of the multi-component intervention proceeded deductively by two members of the research team. Meaning units (quotes) were grouped under their respective constructs (main categories) and summaries were drafted. A second coder reviewed the summaries and compared them to the data. Differences were resolved through peer debriefing [60]. Coding of participants' recommendations for intervention improvement proceeded inductively, allowing for new main categories to emerge. The final coding scheme consisted of 21 codes with 95 incidences. Agreement for coding themes were substantial (K = 0.92). The constant comparative method was used to identify patterns in the data and refine the categories [61]. We used the Consolidated Criteria for Reporting Qualitative Research guideline statement to assist in the reporting of the study [62].

## Results

### 1. Participant characteristics

Most participants were 40–59 years old (79.7%), non-Hispanic black (67.8%), adherent to ART (86.4%), with income below $10,000 (67.8%), and on a federal health insurance program (Medicaid, Medicare) (88.1%) (Table 2). Half of the participants had high nicotine dependence, 76.2% had tried to quit smoking in the last year, and 44.1% reported being depressed (Table 3). In terms of smoking, 59.3% reported current cigar/cigarillo use, 37.3% e-cigarettes, 20.3% chewing tobacco, and 17.0% waterpipe. Overall, 61.9% considered it easy to learn a new app, but only 8.5% had used a health-related app before.

### 2. Focus group themes

Key qualitative findings from focus groups are described below with illustrative comments from focus group participants. A summary of all themes and subthemes are listed in Tables 4 and 5.

**Table 2. Participants demographic characteristics, adherence to HIV treatment, and use of smartphone apps.**

|  | All (n = 59) | Male (n = 31) | Female (n = 28) |
|---|---|---|---|
|  | n (%) | n (%) | n (%) |
| **All** | 59 (100.0) | 31 (100.0) | 28 (100.0) |
| **Age (years)** |  |  |  |
| 18–39 | 2 (3.4) | 1 (3.2) | 1 (3.6) |
| 40–59 | 47 (79.7) | 24 (77.4) | 23 (82.1) |
| >60 | 10 (16.9) | 6 (19.4) | 4 (14.3) |
| **Race/Ethnicity** |  |  |  |
| Non-Hispanic White | 7 (11.9) | 5 (16.1) | 2 (7.1) |
| Non-Hispanic Black | 40 (67.8) | 22 (71.0) | 18 (64.3) |
| Hispanic | 11 (18.6) | 3 (9.7) | 8 (28.6) |
| **Sexual orientation** |  |  |  |
| Heterosexual | 47 (79.7) | 25 (80.7) | 22 (78.6) |
| Gay | 12 (20.3) | 6 (19.4) | 6 (21.4) |
| **Education** |  |  |  |
| Less than high school | 33 (55.9) | 15 (48.4) | 18 (64.3) |
| High school | 14 (23.7) | 7 (22.6) | 7 (25.0) |
| Some college or more | 12 (20.3) | 9 (29.0) | 3 (10.7) |
| **Marital Status** |  |  |  |
| Married/Living with partner | 11 (18.6) | 5 (16.3) | 6 (21.4) |
| Divorced/Widowed/Separated | 15 (25.4) | 7 (22.6) | 8 (28.6) |
| Never Married | 33 (55.9) | 19 (61.3) | 14 (50.0) |
| **Total household income** |  |  |  |
| Under $10,000 | 40 (67.8) | 20 (64.5) | 20 (71.4) |
| $10,000 - $50,000 | 14 (23.7) | 8 (25.8) | 6 (21.4) |
| More than $50,000 | 1 (1.7) | 1 (3.2) | - |
| **Employment** |  |  |  |
| Disabled | 16 (27.1) | 6 (19.4) | 10 (35.7) |
| Employed | 6 (10.2) | 4 (12.9) | 2 (7.1) |
| Not employed | 35 (59.3) | 20 (64.5) | 15 (53.6) |
| **Health care insurance** |  |  |  |
| Uninsured | 3 (5.1) | 2 (6.5) | 1 (3.6) |
| Medicaid | 36 (61.0) | 16 (15.6) | 20 (71.4) |
| Medicare | 16 (27.1) | 10 (32.3) | 6 (21.4) |
| Obama care, employer insurance | 1 (1.7) | 1 (3.2) | - |
| Private/Self | 1 (1.7) | 1 (3.2) | - |
| **Usual health care provider** |  |  |  |
| Private Doctor | 15 (25.4) | 7 (22.6) | 8 (28.6) |
| Community Health Clinic | 16 (27.1) | 6 (19.4) | 10 (35.7) |
| Hospital-Based Health Clinic | 21 (35.6) | 11 (35.5) | 10 (35.7) |
| Emergency Room | 1 (1.7) | 1 (3.2) | - |
| Other | 5 (8.5) | 5 (8.5) | - |
| **Adherence to ART**[*] | 7 (11.9) | 5 (16.1) | 2 (7.1) |
| **Use of smartphone apps** |  |  |  |
| Ever used a health-related app (Yes) | 5 (8.5) | 2 (6.5) | 3 (10.7) |
| It is easy to learn a new app (Yes) | 36 (61.0) | 21 (67.7) | 15 (53.6) |

[*]Suboptimal adherence is defined as reporting <90% adherence to ART in the past 30 days

**Table 3. Participants smoking behavior, alcohol use, and depression.**

| | All (n = 59) | Male (n = 31) | Female (n = 28) |
|---|---|---|---|
| | N (%) | N (%) | N (%) |
| **Believe that smoking puts you at risk (Yes)** | 53 (89.8) | 28 (90.3) | 25 (89.3) |
| **Have a friend who smokes (Yes)** | 54 (92.6) | 28 (90.3) | 26 (92.8 |
| **Tried to quit smoking in the past 12 months** | | | |
| Never | 13 (22.0) | 5 (16.1) | 8 (28.6) |
| 1–5 Times | 34 (57.6) | 19 (61.3) | 15 (53.6) |
| >5 Times | 11 (18.6) | 7 (22.6) | 4 (14.3) |
| **Number of successful quit attempts[a]** | | | |
| Never | 28 (47.5) | 14 (45.2) | 14 (50.0) |
| 1–3 Times | 20 (33.9) | 11 (35.5) | 9 (32.1) |
| >3 Times | 8 (13.6) | 5 (16.1) | 3 (10.7) |
| **Main reason to try to quit smoking** | | | |
| Advice of physician | 11 (18.6) | 7 (22.6) | 4 (14.3) |
| Health reasons, self-initiated | 24 (40.7) | 14 (45.2) | 10 (35.7) |
| The cost | 4 (6.8) | 2 (6.5) | 2 (7.1) |
| Pressure from family or friends | 3 (5.1) | 2 (6.5) | 1 (3.6) |
| Other | 2 (3.4) | - | 2 (7.2) |
| **Smoking cessation treatment received in the past 12 months** | | | |
| None | 39 (66.1) | 19 (61.3) | 20 (71.4) |
| Nicotine replacement therapy | 18 (30.5) | 10 (32.3) | 8 (28.6) |
| Switching to e-cigarettes | 2 (3.4) | 1 (3.2) | 1 (3.6) |
| Individual counseling with Pharmacologic treatment | 1 (1.7) | 1 (3.2) | - |
| **Ever used other forms of tobacco** | | | |
| E-Cigarettes | 22 (37.3) | 11 (35.5) | 11 (39.3) |
| Chewing tobacco (snuff) | 12 (20.3) | 10 (32.3) | 2 (7.1) |
| Cigars, cigarillos | 35 (59.3) | 22 (71.0) | 13 (46.4) |
| Water Pipe (Hookah) | 10 (17.0) | 5 (16.1) | 5 (17.9) |
| **Current use of other forms of tobacco[b]** | | | |
| E-Cigarettes | 6 (10.2) | 4 (12.9) | 2 (7.1) |
| Chewing tobacco (snuff) | 1 (1.69) | 1 (3.23) | - |
| Cigars, cigarillos | 13 (22.0) | 10 (32.3) | 3 (10.7) |
| Water Pipe (Hookah) | 10 (17.0) | 5 (16.1) | 5 (17.9) |
| | Mean (SD) | Mean (SD) | Mean (SD) |
| **Age when started smoking** | 18.2 (8.9) | 16.7 (4.6) | 19.9 (11.8) |
| **Number of cigarettes smoked a day** | 12.1 (7.0) | 13.6 (8.0) | 10.4 (5.3) |
| **Expired CO[c]** | 14.8 (10.4) | 18.2 (12.6) | 11.0 (5.4) |
| **Motivation to quit smoking—Mean (SD)** | 7.0 (2.8) | 7.1 (2.9) | 6.8 (2.7) |
| **Confidence in quitting—Mean (SD)** | 6.9 (2.9) | 6.9 (3.1) | 6.9 (2.7) |
| **High nicotine dependence[d]** | 31 (52.5) | 14 (45.2) | 17 (60.7) |
| **Self-Efficacy/Temptation (Overall)** | 2.8 (0.8) | 2.6 (0.9) | 3.1 (0.7) |
| Positive Affect/Social Situation | 2.6 (1.1) | 2.5 (1.1) | 2.8 (1.0) |
| Negative Affect Situations | 3.1 (1.0) | 2.8 (1.1) | 3.4 (0.6) |
| Habitual/Craving Situation | 2.7 (1.0) | 2.4 (1.1) | 3.0 (0.8) |
| **Risky alcohol use (Yes)[e]** | 18 (30.5) | 11 (35.5) | 7 (25.0) |

(*Continued*)

**Table 3.** (Continued)

|  | All (n = 59) | Male (n = 31) | Female (n = 28) |
|---|---|---|---|
|  | N (%) | N (%) | N (%) |
| **Depressed[f]** | 26 (44.1) | 19 (61.3) | 7 (25.0) |

[a]Tried to quit and succeeded in going without a cigarette for at least 24 hours in the past year

[b]Current use of other forms of tobacco (in the past 30 days)

[c]COppm = carbon monoxide parts per million, a measure of recent smoking obtained through an exhaled breath

[d]High nicotine dependence based on Fagerström Test score > 6

[e]In men, a score of 4 or more is considered positive, optimal for identifying hazardous drinking or active alcohol use disorders. In women, a score of 3 or more is considered positive

[f]Depressed based on CES-D-10 Score ≥ 10.

**Barriers to quitting smoking.** Participants in our focus groups were consistent in their description of emotional stress as the primary reason for failing to quit: "I stopped, then I started again because my mother passed away" (female) (Table 4). Many participants discussed smoking as a means of coping with general life stressors such as feeling lonely: "I moved to Puerto Rico and I found myself by myself. So, all I had to do is smoke and just smoke and smoke" (female) or losing a loved one: "I stopped. Then I started again because my mother passed away. On a good day, I don't smoke. On a bad day, it's a pack" (female). Other barriers to quitting smoking were being around many smokers: "I'm always around smokers at my job" (femal), and nicotine addiction: "I need a cigarette in the morning. This is my worst part of smoking" (male).

Participants stated that their health care providers always advised them to quit smoking given their serious health conditions (e.g., asthma, heart disease). However, participants reported that real assistance in quitting smoking was never discussed or provided "When you go to the doctor, they ask you if you smoke, and they just tell you, 'You know, you have to stop, it's not good for your health.' But they never tell you, 'Okay, go to this program, it will help. I'm referring you–I'm demanding you to go to that to help you, they don't do that!'" (male). Most participants reported using NRT on their own without being advised by health-care professionals. Most of those who used NRT reported negative experiences due to having side effects: "After two or three days, I get the palpitations from the patches" (female). Participants also felt that NRT use was not helpful in reducing their craving to smoke and stated that the patches were very expensive and not affordable: "The patches were expensive more than cigarettes. So, I just went and bought cigarettes" (female).

Another barrier to quitting smoking was multiple drug use addiction. For example, some participants reported that drinking alcohol and smoking go together as a conditioned association: "I had to have both" (female). Others reported starting or increasing their cigarette smoking as a substitute after stopping drug/alcohol use: "When I stopped drinking and drugging, that is when I started smoking more. So, I am substituting the drug for the cigarettes" (female). In addition, participants mentioned that tobacco use was never addressed during their alcohol/drug use treatment, and they felt disappointed they were not advised about the risk of smoking: "In the treatment program, whether you are a drug addict or an alcoholic, they are only teaching you to deal with that, that drug addict, that issue. But you can even smoke in there" (female).

**B. Perceptions about the value of the app.** Participants indicated a high interest in using the app technology to support smoking cessation: "I think the app is really helpful. It will motivate me (Table 5). It will support me. A lot of times I get frustrated" (female). For example,

**Table 4. Summary of identified themes in focus group with examples of participants' comments.**

**Barriers to quitting cigarette smoking**

| | |
|---|---|
| 1. Using nicotine replacement treatment | • I tried the gum; the gum was really nasty. It gave me the hiccups (FG7-F2).<br>• When I first started chewing them, it burnt my mouth (FG7-F7).<br>• After two or three days, I get the palpitations from the patches (FG4-F6).<br>• It did not do anything for me. At first, it might kick it back a little. But I could have a patch on with 21 milligrams, a piece of gum in my mouth, and still smoking (FG3-M8). |
| 2. Low access to tobacco treatment | • I never received treatment. When I quit smoking, I quit cold turkey (FG3-M1)<br>• When I go and see my doctor every three or four months, he always asks me, have you stopped smoking yet? You need to stop smoking because you are in the age of having a heart attack (FG1- M1). |
| 3. Multiple drug use addiction | • When I stopped drinking and drugging, that is when I started smoking more. So, I am substituting the drug for the cigarettes (FG2-F5).<br>• In the treatment program, whether you are a drug addict or an alcoholic, they are only teaching you to deal with that, that drug addict, that issue. But you can even smoke in there, in the dorms, in a smoke area (FG7-F7).<br>• I had to have both. Now, when I quit drinking alcohol, all that was left was the cigarettes (FG7-F7). |
| 4. Stressful conditions and traumatic life events | • I stopped. Then I started again because my mother passed away. On a good day, I do not smoke. On a bad day, it is a pack (FG2-F4).<br>• I got into an argument with my mom and my dad, and I started back smoking (FG6-M1). |
| 5. Being around other smokers | • I am always around smokers at my job, they smoke, smoke. . . you know? (FG7-F7).<br>• When nightclubbing with my friends, or having parties at the house, everybody drinks and puff-puff, and you know, you got to go, then they are pulling me a cigarette out (FG7-F2) |
| 6. Nicotine addiction and craving | • Cravings. When you feel depressed or you feel anxious or whatever. It has to do with the mental behavior (FG 4-F1).<br>• I can feel it in my nerves, especially if I get pissed off at my kids or something. I have got to light it up. I am like, "Let me light it up before I kill you" (FG4-F6).<br>• I be needing a cigarette in the morning. This is my worst part of smoking, See, if I could get past that morning cigarette, I would probably can (FG8-M3) |

**Contingency management related themes**

| | |
|---|---|
| 1. Familiarity with contingency management | • This is the same thing that the alcohol study is using—and it has proven effective for people who stopped drinking. Because they go for 30 days the first time and if they drink, they lose money. But if they do not drink, they get more money every day. In the end, most of the people end up really quiet drinking (FG3-M7).<br>• Like Weight Watchers. I used to go to Weight Watcher every Saturdays. I lost two pounds and everybody applauses—I felt confidence. And when you get five points, you get a keychain (FG5-F5)<br>• The incentive is definitely a motivator (FG3-M7). |
| 2. Amount and type | • $20.00 to 50.00 each week sounds good (FG5-F6)<br>• I like cash (FG7-F6) |
| 3. Resistance to money reward | • I do not really need no money. I can be proud of myself (FG8-M1).<br>• I do not feel like you owe me anything to help me stop smoking (FG4-F1). |
| 4. Useful for boosting motivation | • Sure. It is something I would do. I feel like I am being rewarded for not smoking (FG4-F1).<br>• The incentive is definitely a motivator (FG3-M7) |
| 5. Verifying smoking status by submitting a video via app | • That would be good because then you would know what you were doing–cutting down on your cigarettes or smoking more (FG5-F2).<br>• Yeah, I think it will work (FG8-M1) |

**Adding tracking for HIV adherence to treatment**

<div align="right">(<em>Continued</em>)</div>

**Table 4.** (Continued)

| 1. Agreement | • Yeah, a reminder will be good. Sometimes I forget (FG8-M2)<br>• Yes, sometimes I forget to take my medications because my I'm so busy (FG5-F6).<br>• Yeah, especially for someone just started taking medication. I wished there was something like that for me. I was trying to find something that would remind me every day to take my medication (FG1-M6).<br>• For people that are newcomers that is a good thing (FG5-F3) |
| --- | --- |
| 2. Disagreement | • I do not need a reminder. I never forget my bills. I take mine every morning when I get up (FG4-F2)<br>• If you lose this phone, then everybody knows your business (FG5-f2). |

participants noted several advantages to receiving the app treatment such as ease of access and use, distraction and replacing habits, minimal time commitment, and the chance to practice MT exercises through video. According to one participant: "You can make friends of the app" (male). The most desired features in the app were goal setting, the ability to reach out to others through text messaging other smokers trying to quit, the potential benefits of the videos' health information, text notifications, and the ability to track smoking.

However, participants expressed several concerns with using the app. Some participants thought that the app was expensive and should be provided for free, or at least provided for free for a month to try it before buying it: "I don't think anybody in here is going to download an app and pay $25.00 a month" (female). Some participants were also concerned given their limited knowledge and experience with using technology, and they were worried that using the app might prevent them from benefiting from receiving support and reinforcement from in-person interaction with their health care providers: "You know I'm old. I don't know too much about that technology. My son could help me" (female).

**C. Perceptions about the MT.**   Most participants stated that they did not know what MT was. However, five women and two men mentioned that sometimes they practiced meditation or yoga, relating these two practices to MT. One man reported that he had previously practiced MT when he was receiving treatment to stop drug use in a research study: "At a drug treatment center, they taught me about MT" (male). Almost all participants thought that MT would be helpful in quitting smoking. Participants felt that practicing MT might help in reducing stress, dealing with difficult situations, and staying: "Yeah, it's something good to practice. It'll keep you with positive thought. Positive head. You can get rid of a lot of negative vibes out your head sitting there breathing" (male).

Participants' reactions to practicing the "RAIN" exercise were mixed. Some participants were unconvinced about the benefit of the exercise and thought that the exercise might even tempt them to smoke instead of preventing it: "For somebody who already smokes–it makes me want to go outside right now and smoke a cigarette" (male). On the other hand, some participants felt disgusted by cigarettes after practicing the exercise and stated that the exercise made them more aware about the negative effect of smoking on their lungs and body: "Being aware that I'm smelling it the spark goes on. What am I getting out of all this is awareness" (male).

Participants felt comfortable, calm, relaxed, and a little sleepy after practicing "body scan." Some participants stated that it was a great distraction from smoking and made them more aware about certain parts of their body, which was something new for them: "I love it. I was stiff and then I've been loosening up" (male). Participants also liked practicing "loving kind-ness" and described it as a new concept for them. The exercise made them feel happy, relaxed, and free of pain: "It was relaxing. Going to happy places and breathe it in. Kind of blow the

**Table 5. Summary of identified themes for participants' perception value of the "Craving to Quit" app in focus group with examples of participants' comments.**

| | |
|---|---|
| **Thoughts and feelings on the mindfulness training provided in the app.** | |
| Knowledge and perception value of MT practices | • I do not really understand MT (FG8-M6)<br>• I have done meditation before; yeah, I have done yoga (FG5-F2).<br>• At a drug treatment center, they taught me about MT (FG3-M3)<br>• MT helps you think about some positive stuff and it will take the craving away. It helps (FG8-M3) |
| RAIN | • To me, it did not do anything. It is just an example of what we go through. We know that (FG2-F3).<br>• The enticing the cigarette itself. For somebody who already smokes–it makes me want to go outside right now and smoke a cigarette (FG1-M2)<br>• Being aware that I am smelling it the spark goes on. What am I getting out of all this is awareness (FG6-M3).<br>• Yeah, it is something good to practice because it'll help you in different fields in life. It will keep you with positive thought. Positive head. You can get rid of a lot of negative vibes out your head sitting there breathing (FG1-M3) |
| Body Scan | • I love it. I was stiff and then I have been loosening up (FG8- M3).<br>• To tell you the truth, I never sit down and meditate like that and try to use my inner feelings to feel certain parts of my body. It was totally new for me. Before we started, I was tense, and in the beginning, I said "I can't go through this, I'm going to go get a cigarette" but as I'm going–following the instructions, I just started feeling relaxed (FG7- F5) |
| Loving Kindness | • Yeah. It's something new to me and a good way of thinking that I never thought before (FG8- M3).<br>• Yeah. It was relaxing. Going to happy places and breathe it in. Kind of blow the pain out (FG8- M1). |
| **Thoughts and feelings on videos and messages provided by the app** | |
| Cost & benefit | • I really thought about it because it hits you in the pocket. I spend over $100 a month. And I think about what I can do with that money (FG2-F5).<br>• $7.50, $8.00. So, you add up all of that, within a month? Talking about $1,000.00 you are spending on cigarettes when I could do something else better (FG5-F4). |
| Tripping on Thoughts | • I liked it when she said you are walking down the same street, but then at the end, walk down a different street. That is like mind changing (FG4-F7).<br>• It brought tears to my eyes. The video was very good. It hit home (FG2-F3) |
| Goals | • I like it! That is a good strategy (FG3-M10).<br>• I am going to think about not smoking instead of just, oh, I am going to light one up. I am going to put it over everything to think about not to smoke. That is what I'm going to try to do (FG3- M10) |
| **Concern about the app** | |
| Cost | • I would pay the $25.00, I would pay the first $25.00 just to try it. But if I do not–if it doesn't do anything, I'm going to call somebody about (FG7-F7).<br>• The app should be for free for the first couple of days—and then, you pay if you want to continue (FG1-M6)<br>• Cannot afford it, I do not have an income (FG7-F6).<br>• I do not think anybody in here is going to download an app and pay $25.00 a month (FG7-F2). |
| Lack of experience in technology | • I will be honest. I do not know how to download no apps (FG1-M2).<br>• You know I am old. I do not know too much about that technology (FG6-F3). |
| **Recommendations for improving the app** | |

*(Continued)*

**Table 5.** (Continued)

| | |
|---|---|
| Add information about the negative effect of smoking on HIV | • It will be good to hear others story. Like for instance, when I see that documentary video about the effect of smoking on lungs and all the organs, that scared me and motivate m to quit smoking (FG6-M3).<br>• When you say that smoking can cause this and this, that will be a good thing to do (FG8-M5).<br>• But if you show me a picture of a smoker lungs in bad condition and say this is what smoking causes, that would be a deterrent (FG1-M3). |
| Add information on cessation medications | • It will help if you include something about what medication are available, how to use it, how it will help you? (FG7-F1) |
| Add in-person group counseling sessions to the app to get more social support | • I prefer to add in person and group treatment beside the app. I feel it helps more. Because you are actually talking to somebody. That motivates more. More than only the app (FG8-M2).<br>• Yeah, group with the app sounds better—you will get support (FG6-M3).<br>• I say I like group and app too. Yeah, because you learn a lot more (FG5-F5). |

pain out. I didn't even think about cigarettes. I didn't ever tried that. You put yourself in a different place" (male).

Most participants found the "cost and benefits of smoking" video very effective in motivating them to quit smoking and think about how much they usually spend on smoking and how much they could save if they stopped smoking: "I really thought about it because it hits you in the pocket. I spend over $100 a month. And I think about what I can do with that money" (female). Participants were very touched by the poem in the "tripping on thoughts" video. They thought that the video provided a strong message that was very relevant to them ("it hit home") and made them think about their triggers to smoke and how they can avoid them: "It brought tears to my eyes. The video was very good. It hit home" (female). Participants also considered the "setting goals" video very important and helpful to start seriously planning to change their habit and setting their own goals: "When I go out of here, this time I'm gonna think about not smoking. I'm gonna put it over everything. That's what I'm gonna try to do" (female).

**D. Perceptions about CM.** Some participants were familiar with CM and thought that it could be useful in boosting the motivation to quit smoking: "The incentive is definitely a motivator" (male). One man stated that he used this strategy when he participated in an alcohol treatment research study, and that receiving incentives to stop using alcohol was very helpful for him. Similarly, one woman mentioned that when she participated in a Weight Watchers program, others' encouragement (e.g., by clapping) when she lost weight was a great motivation for her to lose more weight: "Like Weight Watchers. I used to go to Weight Watcher every Saturdays. I lost two pounds and everybody applauses—I felt confidence. And when you get five points, you get a keychain" (female). Regarding the amount of the reward, most participants felt that receiving $20.00 to $50.00 cash: "I like cash" (female) on a weekly basis would be the best for them: "$20.00 to $100.00 each week sounds good" (female). However, a few participants were resistant to the idea and considered it inappropriate or embarrassing to receive money to quit smoking: "I don't really need no money. I can be proud of myself" (male). All participants considered verifying their smoking status by submitting a video of themselves taking the expired CO test through the app feasible and easy to do. Some participants also stated that they might benefit from taking the test daily by tracking their progress in reducing or stopping smoking: "That'd be good because then you would know what you were doing–cutting down on your cigarettes or smoking more" (female).

**E. Perceptions about self-monitoring adherence to ART.**   Half of the participants were supportive of this strategy and half were not. Those who were supportive thought that this strategy would help them keep track of their medication and that it would be especially good for those who are very busy or just starting to take medication: "Yeah, especially for someone just started taking medication. I wished there was something like that for me. I was trying to find something that would remind me every day to take my medication" (male). Those who were not supportive thought that adding this strategy to the app would require extra commitment and work. Some participants also expressed concerns about confidentiality and the risk of exposing their private information in case their smartphones were stolen: "If you lose this phone, then everybody knows your business" (female).

**F. Recommendations for improvement.**   Participants recommended adding educational information about the negative and harmful effects of smoking on HIV treatment and prognosis, supported by concrete scientific evidence: "Watching a video about the effect of smoking on lungs and all the organs will scare me and motivate me to quit smoking" (male). A few participants thought that fear-inducing messages about the harmful effect of smoking would motivate them to quit (e.g., pictorial health warning messages, testimonial videos from individuals who were harmed by smoking): "If you show me a picture of a smoker lungs in bad condition and say this is what smoking causes, that would be a deterrent" (male). Participants also recommended adding more information about the type, brand, and cost of medications used for smoking cessation along with instructions on how to use them to maximize their benefit and reduce their side effects: "It will help if you include something about what medication are available, how to use it, how it will help you" (female). Almost all participants recommended combining the app with group in-person smoking cessation counseling. Participants felt that this would motivate them more to quit by receiving support from the group and learning from others' experience: "I prefer to add in person and group treatment beside the app. I feel it helps more. Because you're actually talking to somebody. That motivates more. More than only the app" (male).

## Discussion

This study provides in-depth insight from PLWH who smoke on the potential of a novel multi-component smoking cessation intervention app integrating MT, CM, and ART self-monitoring strategies to adapt the app intervention to their needs. Participants reported multiple drug use, coping with traumatic life events, being surrounded by many smokers, having bad experiences with NRT, and lack of access to tobacco treatment as significant challenges to quitting smoking. The Craving-to-Quit app's design, videos, and content (both MT and messages) were viewed as attractive, informative, and effective in motivating quit attempts. Participants felt it is necessary to add information about the harmful effects of smoking for PLWH and about how to use NRT, and to complement the app with in-person group counseling to receive more support. Participants felt that CM and self-monitoring strategies to improve adherence to ART would be supportive for quitting smoking. However, participants raised concerns over the cost of the app and the safety of their information in case their smartphones were stolen. These findings underscore the need for comprehensive approaches and further clinical research to test the feasibility and effect of multiple strategies to improve smoking cessation in PLWH.

Consistent with prior reports, focus group discussions revealed several barriers to quitting smoking among PLWH, including other drug addictions (alcohol, marijuana), stressful conditions (e.g., death in the family, loneliness, HIV health issues), being around other smokers, and high nicotine addiction [63]. To be effective and successful, a smoking cessation treatment

targeting PLWH must take into consideration the unique individual, community, and psycho-social factors of this high-risk group. In addition, targeting these barriers when developing short- and long-term relapse prevention plans is critical [63]. Participants mentioned that tobacco use was never addressed during their alcohol/drug use treatment and that they were disappointed about not being advised about the risks of smoking. Identifying dual substance and tobacco users in HIV care and providing combined interventions that address both of the problems might improve smoking cessation efforts among PLWH [64].

Participants described MT as a new concept that could keep them positive and help them to relax, reduce stress, and calm their mind. However, some participants doubted the benefit of practicing the informal RAIN exercise and felt that it increased their craving and desire to smoke instead of preventing it. MT has reportedly entailed significant "downsides" for begin-ners. Given that MT involves nonjudgmental attention to events occurring in the present moment, practicing MT can be hard and very uncomfortable (e.g., when noticing sensations of craving, pain, or unpleasant emotions). Some studies have shown that the development of acceptance and self-compassion are important mediating variables for the positive effects of MT [65].

Participants noted several advantages to receiving the app treatment such as ease of access and use, distraction and replacing habits, minimal time commitment, and the chance to prac-tice MT exercises through video. The main challenges for using the app were its high cost, lim-ited knowledge about using technology, and limited access to in-person interactions with healthcare providers. Similar findings regarding the cost of an app were reported in other behavioral change apps' research, indicating that cost could be an important barrier to using (mHealth) apps, particularly among low-income people [66]. Most of our participants were on a federal health insurance program. One possibility to reduce the cost of this and other apps could be by integrating the apps into PLWH's health care and make them covered under this insurance program. However, health systems are still in the early stages of integrating mobile apps into care and further research is needed to understand how to best integrate apps into care and how to meet both patient and provider needs [67].

Participants recommended adding educational information about the harmful effects of smoking on HIV treatment and prognosis, supported by concrete evidence and specific exam-ples. Participants thought that adding testimonial videos or pictorial health warning messages to the app would better motivate them to quit. Similar results were reported in a study explor-ing the potential of a smoking cessation app targeting LGBTQ[+] youth and young adults [68]. Those researchers also recommended adding information about the type, brand, cost, use, and side effects of smoking cessation pharmacotherapy to improve its benefit. This is highly impor-tant because medication adherence is an important clinical problem among PLWH. PLWH usually manage several medication schedules at the same time which might make it difficult for them to be adherent to the smoking cessation pharmacotherapy [63]. Finally, consistent with previous research, most of the current study's participants were interested in combining in-person group counseling with the app to be able to exchange experiences, share feelings, and receive more support from the group interaction [68, 69].

Participants were receptive to CM and thought that receiving a $25 to $50 cash or gift card reward on a weekly basis would be helpful. While some participants supported adding strate-gies to improve adherence to ART, others thought that this would require extra work and raised concerns over privacy and the security of their information in case their smartphones were stolen. This concern is in line with other studies investigating PLWH's interest in using mobile health technologies [70, 71]. Without ubiquitous and strong privacy rules, the true promise of apps to transform the quality of health care may be weakened. Recently, several solutions have been recommended to protect app users, such as preventing breaches (e.g.,

enable password, pin, or passphrase on phones before distribution), obtaining consent, and using encryption and authentication [72]. However, more research into measures to effectively minimize the risk to privacy and security in mHealth is needed. In addition, researchers should understand the pool of policies an app needs to follow to ensure the safety, privacy, and security of user information [67].

The strengths of this study include a relatively large sample size for qualitative research and the examination of MT and CM as relatively novel potential strategies for smoking cessation in PLWH. This study also has several limitations. First, as with all qualitative research, these findings are not generalized. However, our sample size was appropriate for qualitative research and data collection continued until data saturation had been reached [73]. Second, participants did not use the app outside of the focus group setting. However, we explored participants' perceptions about MT after their real practicing of several MT exercises during the session. Furthermore, because the data are qualitative, conclusions about the strength of our findings cannot be made. However, our data provided valuable insight into how to adapt the app to be most useful to PLWH.

## Conclusions

In summary, results from this study will inform the adaptation of the "Craving-to-Quit" app based on participants' recommendations and concerns. Further investigation on how to reduce the cost of the app is needed. Most importantly, the app should be secure and private, and include information about the negative effects of smoking on PLWH and about medications used in tobacco treatment. Combining in-person group counseling with the app for PLWH is also warranted.

## Acknowledgments

Study data were collected and managed using REDCap electronic data capture tools hosted at the University of Miami. REDCap (Research Electronic Data Capture) is a secure, web-based software platform designed to support data capture for research studies, providing 1) an intuitive interface for validated data capture; 2) audit trails for tracking data manipulation and export procedures; 3) automated export procedures for seamless data downloads to common statistical packages; and 4) procedures for data integration and interoperability with external sources.

**Declarations:** We declare that the work described here has not been published previously, that it is not under consideration for publication elsewhere, that its publication is approved by all authors and tacitly or explicitly by the responsible authorities where the work was carried out, and that, if accepted, it will not be published elsewhere in the same form, including electronically, in English or in any other language, without the written consent of the copyright-holder.

## Author Contributions

**Conceptualization:** Taghrid Asfar.

**Formal analysis:** Taghrid Asfar.

**Funding acquisition:** Taghrid Asfar.

**Investigation:** Taghrid Asfar.

**Methodology:** Taghrid Asfar.

**Writing – original draft:** Taghrid Asfar, Maria Luisa Alcaide, Deborah L. Jones, Laura A. McClure.

**Writing – review & editing:** Taghrid Asfar, Maria Luisa Alcaide, Deborah L. Jones, Laura A. McClure, Judson Brewer, David J. Lee, Adam Carrico.

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
