## [Decision Letter · Decision Letter 0]

12 Jul 2022

HIV patients’ perceptions of a potential multi-component mindfulness-based smoking cessation smartphone application intervention.

PONE-D-22-14683

Dear Dr. Asfar,

We’re pleased to inform you that your manuscript has been judged scientifically suitable for publication and will be formally accepted for publication once it meets all outstanding technical requirements.

Kind regards,

Yukiko Washio, Ph.D.

Academic Editor

PLOS ONE

Journal Requirements

1. Thank you for stating the following in your Competing Interests section:  

All authors certify that they have NO affiliations with or involvement in any organization or entity with any financial interest (such as honoraria; educational grants; participation in speakers’ bureaus; membership, employment, consultancies, stock ownership, or other equity interest; or expert testimony or patent-licensing arrangements), or non-financial interest (such as personal or professional relationships, affiliations, knowledge or beliefs) in the subject matter or materials discussed in this manuscript.

Reviewers' comments:

Reviewer's Responses to Questions

**Comments to the Author**

1. Is the manuscript technically sound, and do the data support the conclusions?

Reviewer #1: Yes

Reviewer #2: Yes

2. Has the statistical analysis been performed appropriately and rigorously? 

Reviewer #1: Yes

Reviewer #2: Yes

3. Have the authors made all data underlying the findings in their manuscript fully available?

Reviewer #1: No

Reviewer #2: Yes

4. Is the manuscript presented in an intelligible fashion and written in standard English?

Reviewer #1: Yes

Reviewer #2: Yes

5. Review Comments to the Author

Reviewer #1: Overall this is a really well written paper. The methods are sound and the findings were interesting to read. One question I had is how is this app specific to people living with HIV? Couldn’t this app also apply to people who smoke who do not live with HIV?

Reviewer #2: This is a well written manuscript that aptly describes a qualitative study to explore the use of a tobacco cessation app to engage PLWH to quit smoking, reduce their stress and adhere to medications needed for their health and survival. The paper is well organized and presents a complicated app develop process and its components in a method that readers will understand. The MT is conceptualized within the framework of typical mindfulness interventions although it does adapt RAIN and Loving Kindness which, in fact, will likely improve the mindfulness app. The inclusion of CT is novel with MT and therefore pushes the science on the barriers to tobacco cessation among this specific population group. The results are well developed, organized and written and the comment inclusion is useful to the reader.

6. PLOS authors have the option to publish the peer review history of their article (what does this mean?). If published, this will include your full peer review and any attached files.

Reviewer #1: No

Reviewer #2: **Yes: **Diane J. Abatemarco, PhD, MSW

---

## [Editor Report · Acceptance letter]

16 Aug 2022

PONE-D-22-14683 

 HIV patients’ perceptions of a potential multi-component mindfulness-based smoking cessation smartphone application intervention.   

Dear Dr. Asfar:

I'm pleased to inform you that your manuscript has been deemed suitable for publication in PLOS ONE. Congratulations! Your manuscript is now with our production department. 

Kind regards, 

on behalf of

Dr. Yukiko Washio 

Academic Editor

PLOS ONE